

# Superaggregates or instrument artifact?

Ashley M. Pierce[1], S. Marcela Loría-Salazar[2], W. Patrick Arnott[2], Grant C. Edwards[3], Matthieu B. Miller[3], Mae Sexauer Gustin[1]

[1]Department of Natural Resources and Environmental Sciences, University of Nevada Reno, NV, USA 89557
5 [2]Atmospheric Science Program, Department of Physics, University of Nevada Reno, NV, USA 89557
[3]Department of Environmental Sciences, Faculty of Science and Engineering, Macquarie University, Sydney, New South Wales, Australia

*Correspondence to*: Ashley M. Pierce (ash.pie4@gmail.com)

10 **Abstract.** Previous studies have indicated that superaggregates, clusters of aggregates of soot primary particles, can be formed in large-scale turbulent fires. High intensity fires may also produce the right circumstances to inject plumes into the upper troposphere and lower stratosphere during pyrocumulonimbus thunderstorms, where the superaggregates can then be transported long distances. Due to lower effective densities, higher porosity, and lower aerodynamic diameters, superaggregates may be deposited past inlets designed to stop particles < 2.5 µm in aerodynamic diameter (PM$_{2.5}$). Ambient 15 particulate matter samples were collected at Peavine Peak, NV, USA (2515 m) northwest of Reno, NV, USA from June to November 2014. The Teledyne Advanced Pollution Instrumentation (TAPI) 602 Beta$^{Plus}$ particulate monitor was used to collect PM$_{2.5}$ on two filter types. During this time, particles > 2.5 µm in aerodynamic diameter were collected on 36 days. On preliminary analysis, it was thought that these particles were superaggregates, depositing past PM$_{10}$ (particles < 10 µm in aerodynamic diameter) pre-impactors and PM$_{2.5}$ cyclones. However, further analysis revealed that these particles were 20 dissimilar to superaggregates observed in previous studies. To determine if the particles were superaggregates or an instrument artifact, elemental analysis, presence of fires, high relative humidity and wind speeds, as well as the use of generators onsite were investigated. Samples with aggregates were analyzed using a scanning electron microscope for size and shape of the aggregates and energy-dispersive x-ray spectroscopy was used for elemental analysis. It was determined that a sampling artifact associated with sample inlet setup and prolonged, high wind events were the probably reason for the observed 25 aggregates.

## 1 Introduction

When primary particles collide and stick together, agglomerates or aggregates can form, creating complex structures (Kulkarni et al., 2011a). Agglomerate particles can be categorized as branched-chain or compact aggregates (Kulkarni et al., 2011a). Branched-chain particles with internal voids between branches and compact aggregates with internal voids have mass 30 equivalent diameters that are less than the volume equivalent diameter, which implies lower densities than an equivalent ideal spherical particle (Kulkarni et al., 2011a). Soot particles are fractal-like, chain aggregates produced from incomplete





combustion (Kulkarni et al., 2011a;Wang et al., 2017). Large-scale turbulent fires provide vortices where soot aggregates (~100s of monomers) can be trapped in a high particle volume fraction, creating superaggregates consisting of thousands of monomers (Chakrabarty et al., 2014;Kearney and Pierce, 2012;Kulkarni et al., 2011a). Large, turbulent fires can cause pyrocumulonimbus thunderstorm formation, which promote injection of superaggregates into the upper troposphere and lower

stratosphere where they can then be transported long distances (Peterson et al., 2014;Peterson, 2014;Fromm et al., 2010). Superaggregates tend to have larger lengths and mobility diameters than smaller particles; however, they have low aerodynamic diameters (a measure of their terminal settling velocity), lower effective densities, and are more porous, causing different behavior than primary particles or smaller aggregates (Chakrabarty et al., 2014;Kulkarni et al., 2011a).

   Superaggregates were observed, from several wildfires in Chakrabarty et al. (2014) and from a laboratory fire in
Kearney and Pierce (2012), with fractal dimensions ($D_f$) of ~2.6 and lengths of 10 to 20 µm (Chakrabarty et al., 2014;Kearney and Pierce, 2012). These superaggregates had "wispy" or "fluffy" appearances when observed using a scanning electron microscope (SEM, Chakrabarty et al., 2014;Kearney and Pierce, 2012). The elemental composition, using energy-dispersive x-ray spectroscopy (EDS), was found to be mainly carbon and oxygen (Chakrabarty et al., 2014).

   Superaggregates are of concern due to the mobility of the particles. In Chakrabarty (2014), superaggregates were
collected in the third stage of an aerosol impactor with a cut point of < 0.3 µm aerodynamic diameter ($D_a$). Measurement of superaggregates would therefore require different size conventions, beyond the widely used aerodynamic diameter, for detection and measurement (Chakrabarty et al., 2014;Marple and Olson, 2011). From a health perspective, these aerosols could also be deposited deep in the lungs of organisms (John, 2011;Kleinstreuer and Zhang, 2009). The optical properties of the superaggregates are also different, due to the complex morphology, and may contribute 90% more atmospheric warming
compared to a volume-equivalent Mie-sphere (Chakrabarty et al., 2014;Sorensen et al., 2011), requiring models to adjust estimates of climate forcing.

   Ambient particulate matter (PM) samples were collected as part of a project to develop and apply a new particulate monitor configured for the measurement of atmospheric mercury (Hg) and lead (Pb) isotopes (Pierce and Gustin, 2016;Pierce et al., 2017). One Teledyne Advanced Pollution Instrumentation (TAPI) 602 Beta[Plus] particulate monitor was located at Peavine
Peak, NV, USA (PEAV, 2515 m) and another TAPI was located ~12 km southeast in Reno, NV, USA (UNRG, 1367 m), to collect particles < 2.5 µm in aerodynamic diameter ($PM_{2.5}$). During the sampling campaign, when instruments were deployed simultaneously at PEAV and UNRG, June to November of 2014, 36 days had particles > 2.5 µm in aerodynamic diameter on sample filters at PEAV, but these were not observed at the lower elevation site (Table 1, Fig. 1 and 2). A season of drought leading to high intensity wildfires in the Western USA (CA, 2017) resulted in numerous smoke events, and preliminary images
of the filters with SEM seemed to support the hypothesis that these particles were fire-generated superaggregates (Fig. 3). The observation of aggregates that did not conform to the description of superaggregates from previous studies led us to wonder if the observed aggregates may in fact be an artifact of the instrument setup and not an ambient air phenomenon. Possible explanations for the aggregates, including elemental composition, presence of fires and fire indicators, correlations with relative humidity (RH) and wind speed, as well as the use of generators onsite, were investigated. SEM and EDS were used to



determine the shape and elemental composition. This paper presents the results of this investigation and the most probable cause for the observed aggregates.

## 2 Site descriptions

The Peavine Peak, NV, USA (PEAV, 2515 m asl, 39.5895 N, -119.9290 W) measurement site was located above tree
line in a sage/steppe ecosystem at the summit, ~15 km east of the Sierra Nevada Mountain range and ~12 km northwest of downtown Reno, NV, USA. The measurement trailer was located within a fenced area that also contained a radio and cellular relay station. The fence restricted unauthorized visitors from approaching within ~15 m of the measurement trailer. There were weekly visits to the site for maintenance of the relay station. There is one dirt road that leads up to the site from the southwest, all other dirt roads and trails are >500 m away from the site and lower in elevation. Traffic in the area consists of off-road gas
and diesel vehicles (trucks, ATVs, and dirt bikes), as well as non-motorized traffic. Back-up power diesel generators, for the relay station, were periodically operated at the site. During June to October 2014, PEAV was, on average, within the planetary boundary layer from the valley and was influenced by upslope mixing from the valley and free tropospheric air (Pierce et al., 2017). In November 2014, the average planetary boundary layer height was below PEAV and thus, air from the free troposphere primarily influenced the site.

The lower elevation site was located near the valley floor at the University of Nevada, Reno Greenhouse complex (UNRG) at the Nevada Agricultural Experiment Station Greenhouse Facility (1367 m asl, 39.5374 N, -119.8044 W) in Reno, NV, USA near the intersection of two major highways, Interstate-80 and Interstate-580 (U.S. Route 395). UNRG and PEAV are ~1 km different in elevation.

Great Basin National Park, NV (GBNP, 2061 m asl, 39.0052 N, -114.2161 W) is located on the eastern side of Nevada.
Measurements occurred from March to October 2015. The measurement trailer was collocated with a Clean Air Statuses and Trends Network (CASTNET) site.

## 3 Instrumentation and data sources

### 3.1 Particulate measurements

The TAPI was configured to measure $PM_{2.5}$ through two separate inlets each with a 10 µm pre-impactor (FAI
Instrument S.R.L. Fonte Nuova, Rome) and a 2.5 µm cyclone (VSCC-A, BGI inc. Waltham, Ma, USA) in-line to prevent particles < 2.5 µm in aerodynamic diameter from continuing in the sample stream (Fig 4). Ambient air was sampled at 16.7 Lpm for 24 h (00:00 to 00:00 PST) simultaneously through two filter mediums: 47 mm cation exchange membranes (CEM; Pall Corporation, PN: MSTGS3R, Line A), and 47 mm Teflon (Pall Corporation, PN: EW-36329-08, Line B). Particulate matter mass concentration on the filters was measured using β attenuation (Sohirripa Spagnolo, 1987). At 24 h resolution the
TAPI has a detection limit of 0.3 µg m$^{-3}$ (TAPI, 2012). CEM filters were destructively analyzed for total Hg (Pierce and Gustin,



2017), Teflon filters were used for Pb isotope (Pierce et al., 2017) and aggregate analysis. Teflon membranes are made of polytetrafluoroethylene (PTFE), a hydrophobic fluorocarbon.

Inlets were connected to the instrument by 2.1 m (CEM filter line) and 1.7 m (Teflon filter line) anodized aluminum sample tubes (2.54 cm outer diameter), supplied with the instrument, that passed into the temperature-controlled trailer to the

TAPI housed inside (Fig. 4). Inside the temperature-controlled trailer, the sample lines connected to condensation water traps on each line to collect any water droplets that formed on the inside of the sample tubes due to condensation. Just below the condensation water traps were sample line heaters (Fig. 4). The sample line heaters on each line were set to only heat the line if the RH in the sample air stream exceeded 40% RH and would stop heating once the RH reached 30%. A CEM reference filter was also used automatically throughout the sample process to account for humidity effects on the sample filters (TAPI,

2012).

The pressure drop across the CEM filters was higher than the pressure drop across the Teflon filters due to difference in material. The higher-pressure drop on the CEM filters (Pierce and Gustin, 2017;TAPI, 2012) necessitated a different sample line inlet nozzle (located where the sample line enters the TAPI measurement box, Fig. 4), tested and adjusted by Teledyne before the instruments were operated. The inlet nozzle was therefore a different size (smaller in diameter) for the CEM sample

line (0.75 cm diameter) compared to the Teflon sample line (1.9 cm diameter) and may have caused different flow dynamics for the CEM sample line. Constrictions in sample air flow causes gases in the sample stream to increase in velocity and focus in the center of the tube, this dynamic causes increased particle deposition (Kulkarni et al., 2011b). The different inlet nozzle sizes required a different $\beta$ sample area (CEM: 4.7 cm$^2$ $\beta$ sample area, Teflon: 12 cm$^2$ $\beta$ sample area), or the area of the filter used for $\beta$ attenuation. The filters were supported in different filter cartridges for the CEM and Teflon filters due to the

difference in $\beta$ sample area. Filters were automatically loaded and unloaded, and then held in an unloader tube until collection every 1 to 2 weeks. Sample inlets were cleaned monthly, following instructions from the instrument manual, and pumps were rebuilt every 6 to 8 months.

Calibration of the operating flow rate regulation system, $\beta$ source span checks, and pneumatic circuit leak tests automatically occurred at the start of each sampling period for both sample lines. There were three days during the sample

period in October 2014 when data validation was not passing after the automatic tests were performed. These days are discussed in sect. 4.5. On 25 days, of 158 in the sample period, there were warnings related to pump valve, span tests, leak tests, or internal cooling fan failure. These warnings cleared and the sample passed data validation for the day. Six days with warnings occurred on days with aggregates. Four days, June 13, 16, 19, and September 16, were pneumatic leak test warnings; 2 days, September 18 and 23, were pump valve warnings. Aggregates on the June warning days occurred only on the CEM

inlet, aggregates on the September warning days occurred on both inlets.

### 3.2 Shape, size, and elemental composition

Teflon filters were analyzed at Macquarie University in Sydney, Australia using a scanning electron microscope (SEM, JEOL USA Inc. model: 6480 LA, Peabody, MA, USA) for size, shape, and elemental composition. Preliminary SEM



analysis was performed with backscattered electron imaging on un-coated filter segments in low vacuum to avoid charging and breakdown of the aggregates. Elemental analysis using energy-dispersive x-ray spectroscopy (EDS) was also performed with backscattered electron imaging in low vacuum mode. A second SEM analysis with secondary electron imaging, using a different filter segment from the initial SEM, was performed with gold coating, to prevent charging during analysis in high vacuum. The second SEM analysis was a more in depth exploration of the morphology of the aggregates.

### 3.3 Fire indicators using aerosol optical properties

Aerosol Optical Depth (AOD, 440 nm) and Ångström Extinction Exponent (AEE, 440-870 nm) were collected from a Cimel (CE-318) sun photometer used in the AErosol RObotic NETwork, located at the University of Nevada, Reno (UNR) campus on top of a four-story building (Loría-Salazar, 2014;Loría-Salazar et al., 2017). One hour data were collected and averaged for 24 h. AOD is a measure of the columnar aerosol loading and when compared with surface $PM_{2.5}$ measurements can aid in identifying periods of wildfires (Loría-Salazar et al., 2017). AEE is used as a qualitative indicator of particle size; AEE ~ 1 is indicative of coarse mode aerosols (i.e. dust and sea salt) while AEE ≥ 2 is indicative of fine mode aerosols from biomass burning or urban pollution (Eck et al., 1999). Data with AEE >1.8 were flagged as fire periods (Loría-Salazar et al., 2016).

### 3.4 Meteorological data

At PEAV, RH was measured by an HMP45c model Campbell Scientific RH monitor (± 0.2 ℃ and ± 2% RH), wind speed was measured by an RM Young 05305 wind vane (± 0.2 m s$^{-1}$). Wind speed at UNRG was collected from the Western Regional Climate Center. Wind speed at GBNP was collected from the CASTNET site. Hourly meteorological data was used for 1 h max values and 2 h averages.

### 3.5 Generator use

From October 20 (Monday) to October 24 (Friday), multiple diesel generators were operated at PEAV while maintenance was occurring on the power lines at the relay station. A back-up power, diesel generator was also located on site and was operated periodically throughout the sample period. The back-up generator was located ~10 m southeast and around the corner of the relay station building from the measurement trailer. Exact timing of back-up generator use is not available.

## 4 Results

### 4.1 Particulate measurements

Of the 36 days with aggregates > 2.5 µm in aerodynamic diameter at the PEAV site, aggregates were observed on both inlets 22 days, the other 14 days, aggregates were observed on one inlet (Table 1). All single inlet days, except for one,




occurred on the CEM inlet. This may be due to the higher-pressure drop across the CEM filter and higher flow constriction causing different flow dynamics for the CEM sample line. Aggregates, when observed only on a single inlet, had light loading, only two of the 14 single inlet days exceeded the 75$^{th}$ percentile concentration (7.1 µg m$^{-3}$) for the sample period and both of those days had visually high PM$_{2.5}$ loading, a fire flag, and minimal aggregates. Presence of aggregates in some cases were

associated with high PM$_{2.5}$ concentrations, 20 days of the 36 had concentrations >7.1 µg m$^{-3}$; however, due to aerosols from fires, this was not always related to aggregate loading.

This instrument was located at the lower elevation site (UNRG) with another TAPI instrument, before and after it was located at PEAV. Correlation between this instrument and the TAPI instrument located at UNRG was high before (r$^2$ = 0.8, p < 0.05, n = 6) and after (r$^2$ = 0.88, p < 0.05, n = 71) it was located at PEAV, indicating that the two instruments were

operating similarly (Pierce and Gustin, 2017). Furthermore, no aggregates were observed when the TAPI was moved to GBNP from March to October 2015, where the instrument was also impacted by fire plumes (Pierce and Gustin, 2017).

Black aggregates occurred on filters on days when inlets were cleaned and on days before and after cleaning (Fig. 2). Black particles were not observed in the PM$_{10}$ pre-impactor or PM$_{2.5}$ cyclone inlets during routine inlet cleaning; brown dust particulates were observed (Fig. 5). During a thorough cleaning on October 3, 2014, after aggregates had been observed for

multiple days in September, black particulate matter was noticed in the condensation water traps (Fig. 6). After cleaning the condensation water traps and reassembling the inlets, aggregates were again observed on multiple days in October.

### 4.2 Shape, size, and elemental composition

During the second, in depth look at the aggregates using the SEM, it became apparent that the aggregates were not predominantly "fluffy" (Fig. 3) like those observed in Chakrabarty et al. (2014) or Kearney and Pierce (2012), but more

compact and did not resemble chain-aggregates (Fig. 7, Table 1). On eight of the 12 filters analyzed on the SEM, aggregates that were fluffy could be located; however, they were outnumbered by compact aggregates such as those in Fig. 7 and Fig. 8. The fluffy aggregates observed were 10 to 20 µm in length (Fig. 3), compact aggregates observed were 10 to >100 µm in length (Fig. 7).

Blank Teflon filters were 25 to 41% carbon (C), 0 to 8% oxygen (O), and 51 to 75% fluorine (F) identified by EDS.

Filter segments with PM$_{2.5}$ but no aggregates, had a range of elements including C, O, F, sodium (Na), magnesium (Mg), Al, Si, S, Cl, K, Ca, and Fe. For these segments, F (21 to 68%) and C (22 to 61%) with some O (3.4 to 19%) and small amounts of the remaining elements dominated the chemical composition. Aggregate chemical composition on the other hand consisted of C, O, F, Mg, Al, Si, S, Cl, K, Ca, and Cu, dominated by F (11 to 59 %), C (15 to 60%), O (4.6 to 38%), Al (0.19 to 42 %) and small amounts of the remaining elements (Table 3).

### 4.3 Fire indicators using aerosol optical properties

There was a drought in the Western USA from 2012 to 2016 that contributed to dry conditions and many wildfires throughout 2014 (CA, 2017). There were ~52 fires that exceeded 1 km$^2$ of burned area in California and ~63 fires that exceeded



$4 \text{ km}^2$ of burned area in Oregon and Washington in 2014. There were two large fires during the measurement campaign. The first and largest in California for 2014, the Happy Camp Complex Fire, located in northern California, ~400 km northwest of PEAV, burned $543 \text{ km}^2$ from August 14 to October 31 (CA, 2017). The King Fire burned $390 \text{ km}^2$ in El Dorado County, California, ~100 km southwest of PEAV from September 13 to October 9, 2014 (CA, 2017). Several fire plumes throughout the measurement campaign affected PEAV (Pierce et al., 2017), and aggregates occurred more frequently in September 2014 when the fire plume from the King Fire was heavily impacting the area (Fig. 2).

Aerosol Optical Depth (AOD, at 440 nm), a measure of the columnar aerosol loading, was positively correlated with $PM_{2.5}$ at PEAV for all data ($r^2 = 0.33$, $p < 0.05$, Fig. 9a) and higher if only days with aggregates are used ($r^2 = 0.49$, $p < 0.05$, Fig. 9b). If aggregate days were removed, the correlation increased to 0.58 for all data. The positive correlation for AOD and $PM_{2.5}$ for aggregates days was influenced by one point with AOD > 0.8 (highest AOD observed for the entire sample period) that occurred on September 18, when smoke from the King Fire was heavily influencing the sites (Fig. 9b). When that point is removed the correlation decreased to 0.36.

AEE was used here as a general indicator of particle size to identify biomass burning in the region (Fig. 10). For 31 aggregate days with AEE available, 24 days occur during days with fires (Fig. 10b). Of 158 days during the sample period with data from the Cimel, 120 days had fire flags. There was no correlation between AEE and all $PM_{2.5}$ nor between AEE and days with aggregates.

## 4.4 Meteorological data

Ambient RH was measured at PEAV, as well as inside the instrument box of the TAPI during sample collection and sample analysis with β attenuation (Fig. 11). Same day RH was not correlated with $PM_{2.5}$ for all data or days with aggregates for any of the RH measurements (outside RH, internal TAPI RH during sample collection, and internal TAPI RH during sample analysis). The in-line heaters did not turn on during this time, as RH during sampling (Fig. 11c and 11d) did not exceed 40% RH. RH also did not exceed 40% during sample analysis with β attenuation (Fig. 11e and 11f). It is possible that the RH effect on hygroscopic growth of particles was lagged and same day RH would therefore not be an effective measure. However, lagging the RH by 1, 2, and 5 days did not improve the correlation with aggregate days.

Higher wind speeds were observed at PEAV (median: 2.5 m s$^{-1}$, range: 0.0 to 36 m s$^{-1}$) relative to UNRG (median: 1.4 m s$^{-1}$, range: 0.0 to 9.6 m s$^{-1}$). PEAV also had higher wind speeds than GBNP (median: 2.4 m s$^{-1}$, range: 0.0 to 13 m s$^{-1}$) where the TAPI was later located during the fire season of 2015 with no aggregates observed. Wind speed was weakly, positively correlated with $PM_{2.5}$ for all data ($r^2 = 0.33$, p-value < 0.05, n= 152, Fig. 12a). This correlation increased only slightly when only days with aggregates were used in the analyses ($r^2 = 0.39$, p-values < 0.05, n=36, Fig. 12b). Maximum (max) 1 h wind speed was also only weakly correlated with $PM_{2.5}$ ($r^2 = 0.26$, p-value < 0.05, Fig. 12c) for all data and analyses using only days with aggregates ($r^2 = 0.25$, p-value < 0.05, Fig. 12d).

There were 49 days out of 155 at PEAV with wind speed measurements, when wind speed exceeded 10 m s$^{-1}$ for at least one hour of the day. Twenty-seven out of 36 days with aggregates occurred on days with hourly wind speeds >10 m s$^{-1}$



and 33 out of 36 days with aggregates occurred on days with wind speeds >5 m s$^{-1}$. Of the nine days with observed aggregates that did not occur on high wind days, seven days aggregates only occurred on one inlet, which always had light loading. On the 13 aggregate days labelled as medium or heavy loading (Table 1), wind speeds exceeded 20 m s$^{-1}$ for at least one hour of the day or wind speeds exceeded 10 m s$^{-1}$ for 10 h or longer leading up to or during that day. Five days with high wind speed occurred during the week of generator use (October 20 to 24), and it is unclear if aggregates were present due to the high loading on these filters. Of the remaining 17 days with high wind speeds, 12 days occurred either before or after an aggregate day or had only one hour of the day that exceeded 10 m s$^{-1}$.

### 4.5 Generator use

From October 20 (Monday) to October 24 (Friday), multiple generators were operated. On October 22 to 24, the data from the TAPI located at PEAV did not pass data validation due to two errors related to flow rate and pump valve. The heavy loading observed on the filters (Fig. 13a) caused the TAPI to be unable to sample through these filters for the full sample period. This loading quickly cleared from the sample line once the generators were removed (Fig. 13b). A back-up power generator was located on site and was operated periodically throughout the sample period. We do not have exact timing for generator use but most likely, the generator would have been operated on weekdays when maintenance was performed. Aggregates occurred on weekends five times (both inlets 3 days and on one inlet 2 days).

### 5 Discussion

Potential causes of the particles > 2.5 µm in aerodynamic diameter investigated here include presence of fires, high RH potentially causing hygroscopic growth of particulates in the sample stream, high wind events causing degradation of the Al tubing, and exhaust from generators operated on site. The large number of fires in the Western USA and days with fire flags (120 days out of 158 with data available) throughout the measurement campaign and the increase in aggregates during September 2014 when the fire plume from the King Fire was heavily impacting the area (Fig. 2) seemed to support superaggregates generated by high intensity fires. The moderate correlation between AOD and PM$_{2.5}$ (r$^2$ = 0.33 for all data and r$^2$ = 0.49 for aggregate days, Fig. 9) indicated that high aerosol loading in the atmospheric column may influence the occurrence of aggregates. If aggregate days were removed, however, the correlation increased to 0.58 for all data. The removal of the one point with AOD > 0.8 decreased the correlation for aggregate days to 0.36. AEE, used here as a general indicator of particle size to identify biomass burning in the region (Fig. 10), demonstrated that many days with fires occur with no aggregates. Based on findings in Loría-Salazar et al. (2017), we would expect PM$_{2.5}$ and AOD to be positively correlated in certain conditions (unstable conditions in a well-mixed boundary layer and during wildfires). Given the high frequency of fires, if the aggregates were generated and transported in fire plumes, we would expect the correlation with AOD and AEE to be higher.

It is possible that high RH could promote hygroscopic growth of aerosols in ambient air or in the sample stream. Hygroscopic growth factors (diameter of a particle at a certain RH/dry diameter) are estimated for ammonium sulfate,





ammonium nitrate, and sea salt aerosols, for use in the national network, IMPROVE light extinction algorithm (Pitchford et al., 2007). Pure ammonium sulfate crystallizes at 37% RH, and it is assumed no hygroscopic growth occurs below 37% RH, based on the efflorescence or hysteresis branch of the ammonium sulfate growth curve (Pitchford et al., 2007;Clegg et al., 1998). Sea salt aerosols are assumed to have no hygroscopic growth below 47% RH and ambient organic mass particulates are

5 assumed to have limited to no hygroscopic growth (Pitchford et al., 2007).

The Western USA generally has higher ambient organic carbonaceous mass particulates, lower mass concentration of inorganic species known to impact hygroscopic growth (sulfates and nitrates), and strong seasonal fluctuations in boundary layer RH compared to the Eastern USA (Nguyen et al., 2016;Buseck and Schwartz, 2003;Malm et al., 2011;Malm et al., 2004;Malm and Sisler, 2000). Wintertime build-up of particulate nitrate has been observed in Western USA valleys (Green et

al., 2015), but PEAV would likely not be affected by this build-up due to elevation and sample period. Furthermore, Loría-Salazar et al. (2017) found no correlation between AOD measurements and RH in the boundary layer in Reno, NV for a yearlong sample period in 2013. Outside RH did at times exceed 37%; however, internal TAPI RH measurements indicated that the RH in the sample stream did not exceed 37%. Due to the higher ambient organic mass particulates in the West and RH < 37% in the sample line, it does not seem likely that this was the cause of the aggregates.

High wind speed may have caused sections of anodized Al tubing to rub together, also known as fretting corrosion (Davis, 1999;Waterhouse, 1992). Fretting is defined as small-amplitude movement that can occur between contacting surfaces, usually due to external vibration (Waterhouse, 1992). Fretting corrosion arises when dry oxidation during rubbing occurs, producing a black powder of aluminum oxide, more likely to occur when aluminum contacts aluminum (Davis, 1999). Aluminum tubing was in contact with another surface in four places: where the sample tubes connected to the base of the $PM_{2.5}$

cyclones, before and after the water condensation water traps, and where the sample tubes entered the TAPI measurement box (marked in red, Fig. 4). On several occasions during site visits, it was noticed that the $PM_{10}$ and $PM_{2.5}$ inlets were in different positions from the last site visit, indicating that high wind speeds had caused the inlets to rotate on the sample tubes. The observation of a black powder in the condensation water traps (Fig. 6) and not in the $PM_{10}$ pre-impactor (Fig. 5) or $PM_{2.5}$ cyclone supports the generation of aggregates in the sample line after the inlets and before the condensation water traps. The

increase of Al and O in the aggregate samples indicated that the anodized coating on the sample lines was possibly undergoing fretting corrosion. Anodized Al coatings are ~80% aluminium oxide, ~18% aluminium sulfate, and ~2% water.

The samples from October 20 (Monday) to October 24 (Friday), when multiple generators were running at PEAV indicate that aggregates may have been caused by generator exhaust, due to the similarity of the filter at the end of the period to other aggregate filters. Smits et al. (2012) measured generator exhaust emissions at different loadings for a small-scale

generator using a low-sulfur fuel. Diesel generators emit nitrogen oxides (NOx), volatile organic compounds (VOCs), carbon monoxide (CO), and PM (Smits et al., 2012). Elements found in the generator exhaust included potassium (K), calcium (Ca), titanium (Ti), strontium (Sr), chromium (Cr), iron (Fe), nickel (Ni), manganese (Mn), copper (Cu), zinc (Zn), lead (Pb), sulfur (S), and chlorine (Cl). Elements that were not detected included silicon (Si), vanadium (V), selenium (Se), cadmium (Cd), antimony (Sb), and aluminum (Al). SEM from Smits et al. (c.f. Fig. 4 of Smits et al., 2012) look similar to the SEM we have




collected in this study, however, the absence of Al in Smits et al. (2012) differs from our findings (Table 2 and 3). Aluminum was assumed to be a tracer for environmental atmospheric samples in Smits et al. (2012), filters with ambient $PM_{2.5}$ from PEAV are consistent with this assumption with small amounts of Al (Table 2). Aggregate samples however had much higher Al (Table 3), indicating these samples were not background environmental $PM_{2.5}$. Aluminum is not a common additive to diesel fuel or lubricating oil. Aggregates on weekends, when it is unlikely that maintenance was occurring, and the large percentage of Al present indicated that generator exhaust was likely not the cause of the aggregates.

## 6 Conclusions

During the sampling campaign at PEAV from June to November 2014, the presence of fires and fire indicators, high RH, high wind speeds, and use of generators onsite were investigated to understand the presence of ambient particles that exceeded 2.5 µm in aerodynamic diameter on 36 measurement days. Particles >2.5 µm in aerodynamic diameter were not observed on samples from the same days at the lower elevation site, from the TAPI before and after it was deployed at PEAV, nor when it was moved to another high elevation site, GBNP impacted by fires in 2015. RH and AEE were not correlated with aggregate measurements, indicating RH and indicators of fire did not predict aggregate formation. The presence of aerosols in the column (AOD) was correlated with $PM_{2.5}$, however, the positive correlation was heavily influenced by one data point and AOD did not fully explain the presence of aggregates. Linear regression may not be adequate to identify the cause due to mass concentration of the filters being a combination of $PM_{2.5}$ and aggregate mass. Generator use at the site could be the source of the aggregates and would be an interesting area of further research if particulates from the exhaust were able to deposit past the $PM_{10}$ pre-impactor and $PM_{2.5}$ cyclones, however chemical composition did not support generator exhaust as the source of aggregates.

High concentrations of Al and O observed in the EDS elemental analysis suggested that the anodized Al sample tube coating contributed aggregates to the samples. Fretting corrosion, occurring where two sections of Al tubing were rubbing together, caused by prolonged, high wind events at PEAV seems to be the most likely explanation of the observed aggregates. Aluminum tubing was in contact with another surface in four places: the base of the $PM_{10}$ pre-impactor, before and after the condensation water trap, and where the sample tubes enter the TAPI measurement box (marked in red in Fig 4). The observation of black particles in the condensation water traps (Fig. 6) but not on the $PM_{10}$ pre-impactor plates (Fig. 5) or in the $PM_{2.5}$ cyclones also supports fretting corrosion occurring in the sample line downstream of the $PM_{10}$ pre-impactor plates and the $PM_{2.5}$ cyclones. More experiments to test this theory are needed to understand the specific conditions promoting fretting corrosion.

In other particulate monitors, such as the Beta Attenuation Monitor (BAM-1020), used in regulatory networks, filter tape is automatically advanced, without post-processing. The BAM uses a similar inlet setup to the TAPI. At sites with high wind speeds, if there are sections of Al tubing susceptible to fretting corrosion, a similar situation could occur and go unnoticed





and potentially impact measured $PM_{2.5}$ concentrations. The observations presented suggest inlet configuration is important to consider for sites with high wind events.

**Acknowledgments**

We acknowledge the Nevada Division of Environmental Protection (NDEP), the UNR College of Biotechnology and Natural Resources, and a USDA-HATCH grant (NIFA Accession# NEV05295) for supporting this project. We would like to acknowledge help from Sue Lindsey and Katrina Macsween at Macquarie University in Sydney, New South Wales, Australia for assistance with and use of the scanning electron microscope. Thank you to Dave Metts of High Sierra Communications for allowing access to and support at the Peavine Peak site. Thank you to Kristien King for assistance with data processing.





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



## Figures

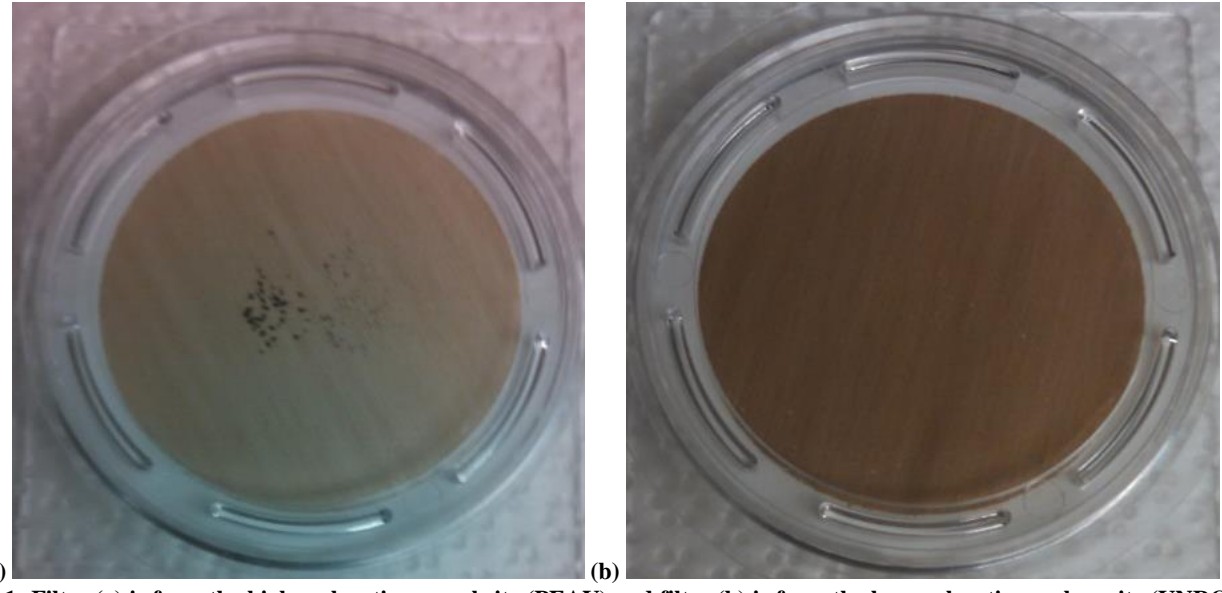

(a)      (b)

**Figure 1:** Filter (a) is from the higher elevation, rural site (PEAV) and filter (b) is from the lower elevation, urban site (UNRG) on September 16, 2014. Aggregates were observed at PEAV but not at UNRG.

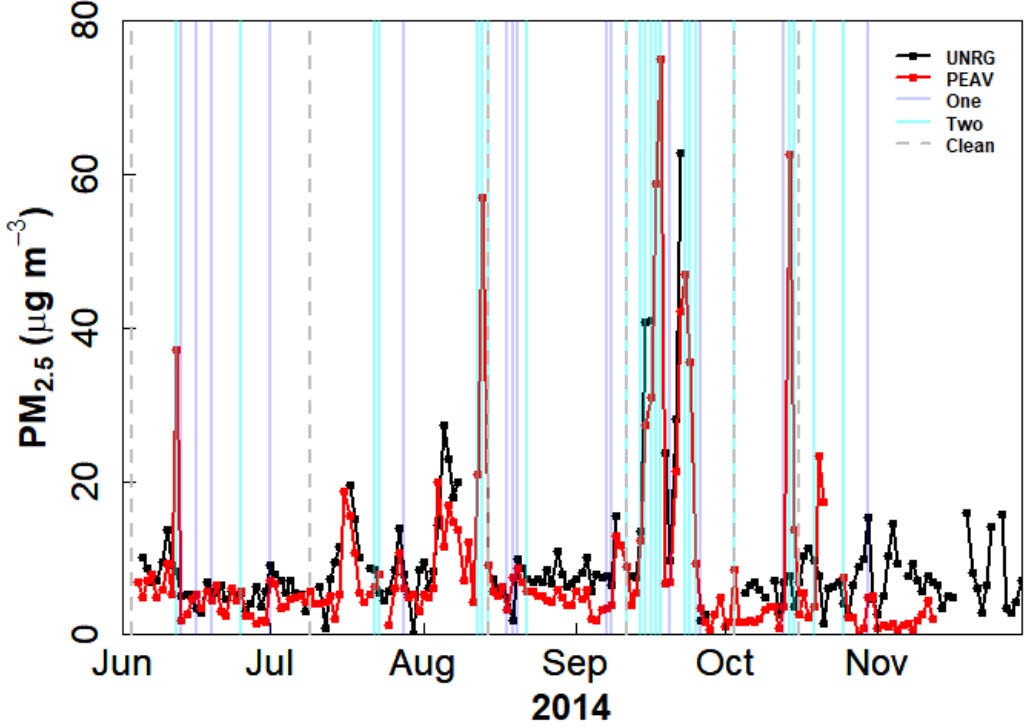



**Figure 2: PM₂.₅ (µg m⁻³) at PEAV (red) and UNRG (black) for June to November 2014. Vertical dark blue lines are days with aggregates on one inlet, vertical light blue lines are days with aggregates on two inlets, dashed grey lines are days when the inlets were cleaned.**

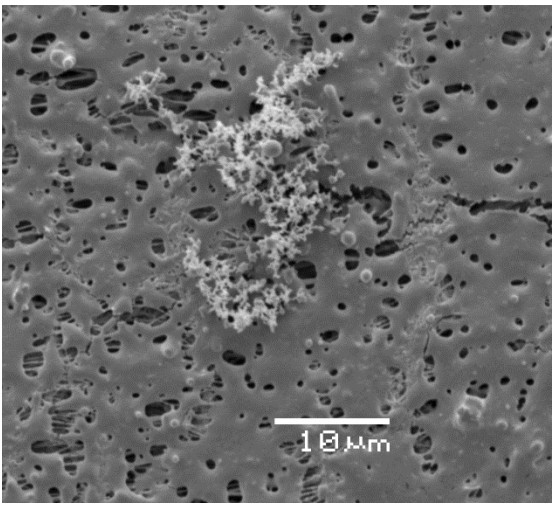

**Figure 3: SEM (uncoated, backscattered electron imaging, low vacuum) image of an aggregate on a Teflon filter from the higher elevation, rural site (PEAV) on September 18, 2014.**



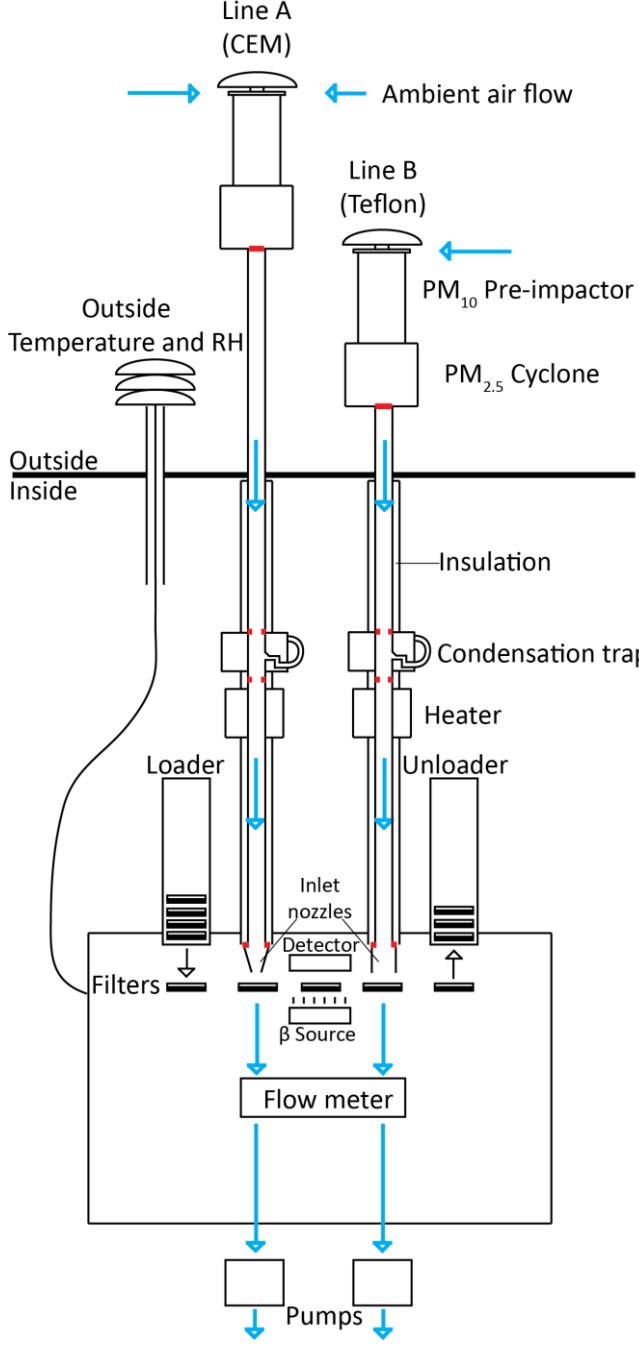

Figure 4: Diagram of Teledyne Advanced Pollution Instrumentation (TAPI) 602 Beta[Plus] particulate monitor (modified from Pierce and Gustin, 2017). Blue indicates the airflow through the instrument. Red indicates where aluminum tubing is in contact with another surface: the base of the PM$_{2.5}$ cyclones, before and after the condensation water traps, and where the sample tubes entered the TAPI measurement box above the inlet nozzles.


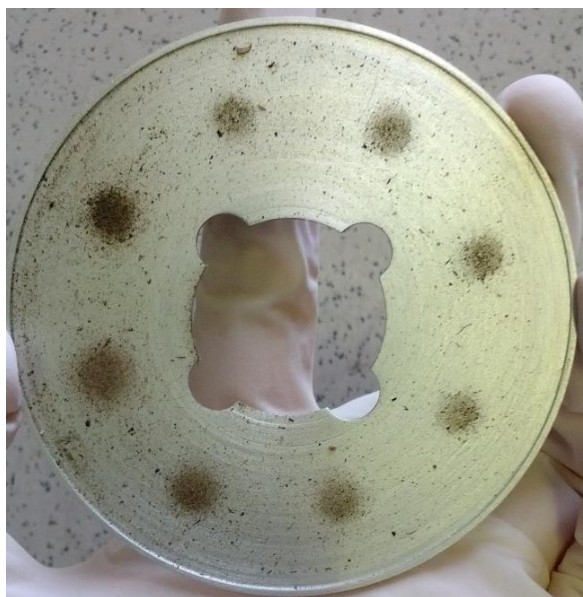

**Figure 5: PM₁₀ pre-impactor plate from the CEM inlet (Line A, Fig. 4) during a routine cleaning after a high wind event.**

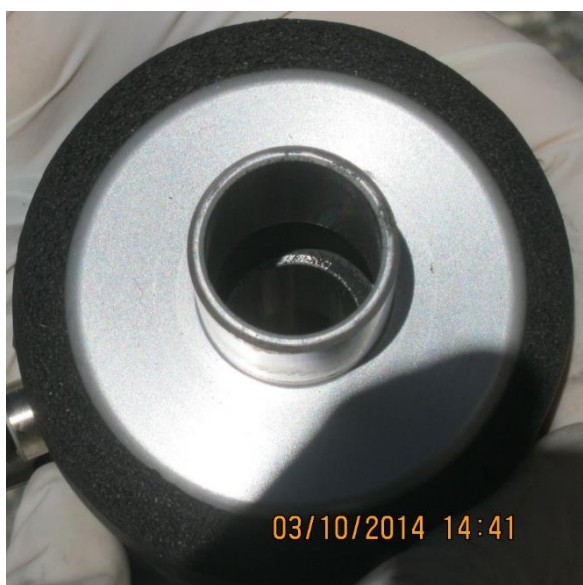

5   **Figure 6: Condensation water trap during a cleaning on October 3, 2014. Black powder was visible in the condensation water trap. The black around the outside of the trap is insulating foam.**





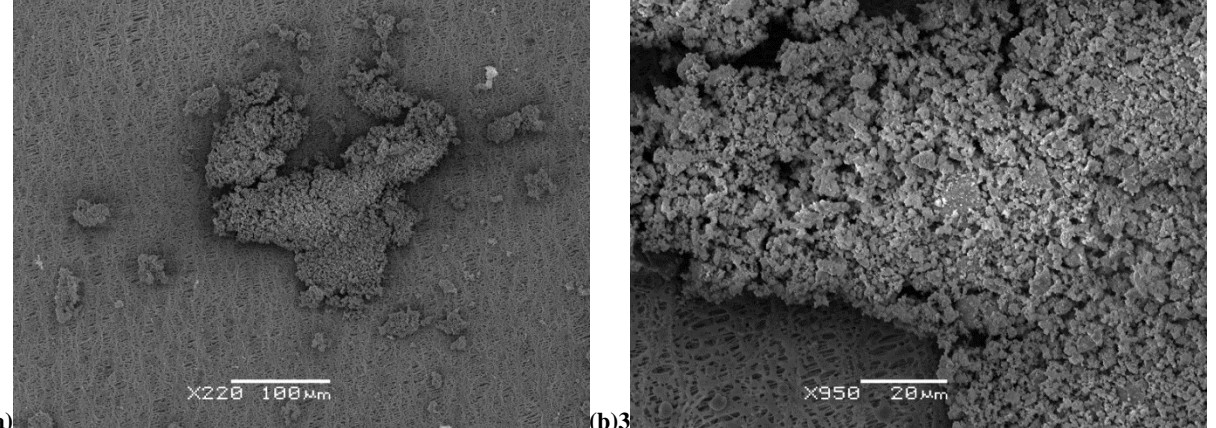

**(a)**                                                            **(b)3**

**Figure 7: SEM images of aggregates (gold-coated, secondary electron imaging, high vacuum) for (a) September 16, 2014 (b) higher magnification of same aggregate.**

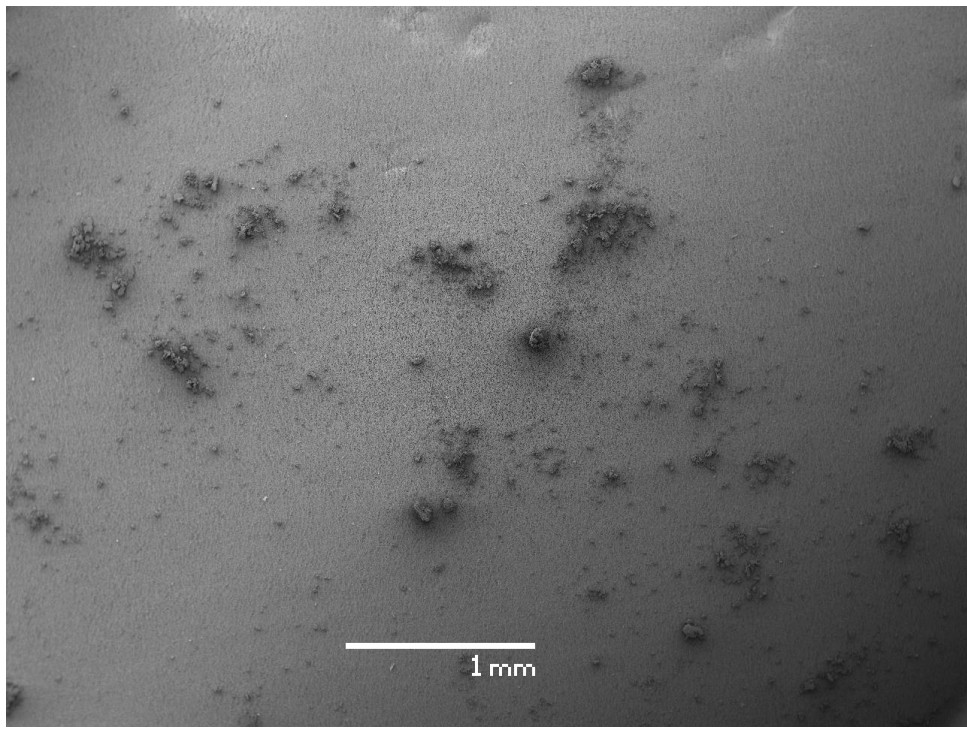

**Figure 8: SEM image (uncoated, backscattered electron imaging, low vacuum) of aggregates on September 16, 2014.**





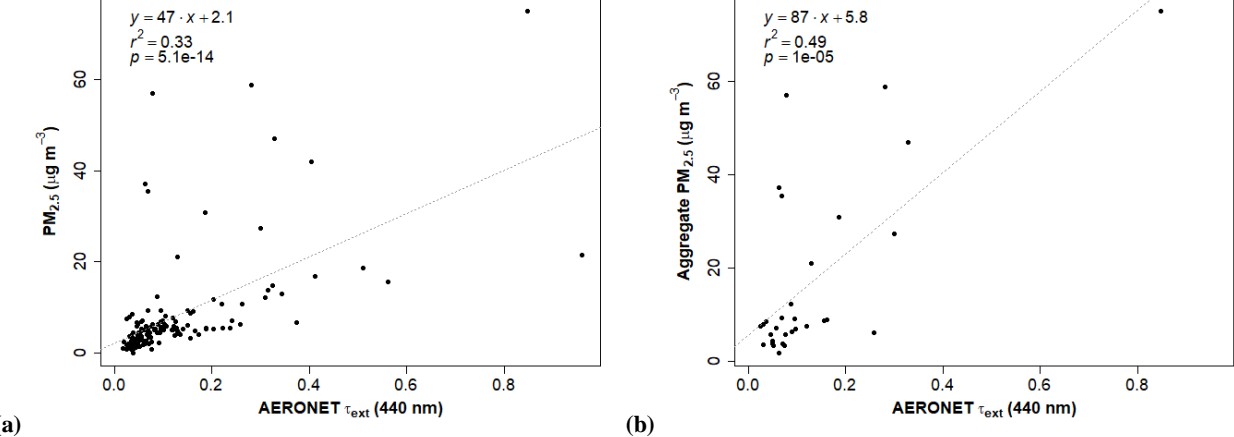

**(a)**                   **(b)**

**Figure 9: Aerosol Optical Depth (AOD, $\tau_{ext}$) from the Cimel sun photometer located in the valley plotted against PM$_{2.5}$ (µg m$^{-3}$) at PEAV for (a) all data and (b) days with aggregates.**

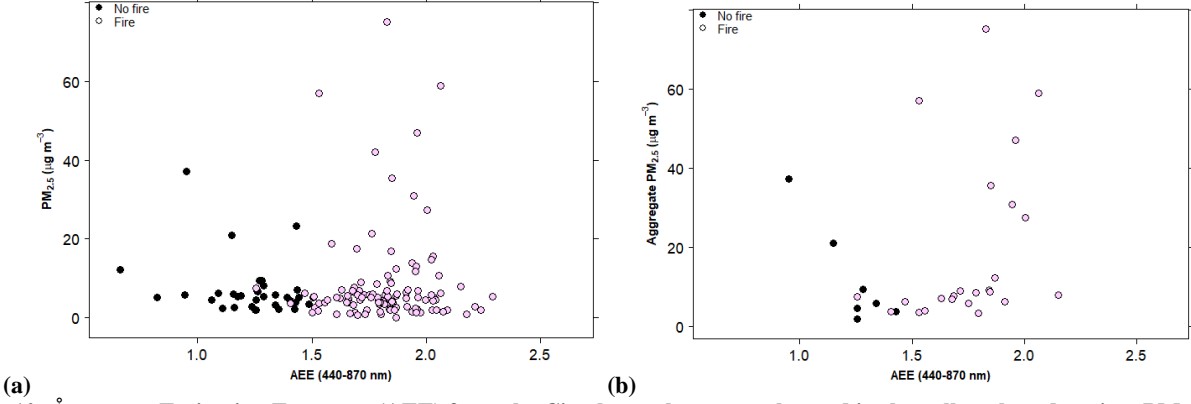

**(a)**                   **(b)**

**Figure 10: Ångström Extinction Exponent (AEE) from the Cimel sun photometer located in the valley plotted against PM$_{2.5}$ (µg m$^{-3}$) at PEAV grouped by days with (open points) and without (filled points) fires for (a) all data and (b) days with aggregates.**





**Figure 11: Relative Humidity (%) and PM$_{2.5}$ (µg m$^{-3}$) measured June to November 2014 for (a) ambient RH for all data, (b) ambient RH on aggregate days, (c) average instrument RH during sampling for all data, (d) average instrument RH during sampling for days with aggregates, (e) average instrument RH during β-attenuation collection for all data, and (f) average instrument RH during β-attenuation collection for days with aggregates.**




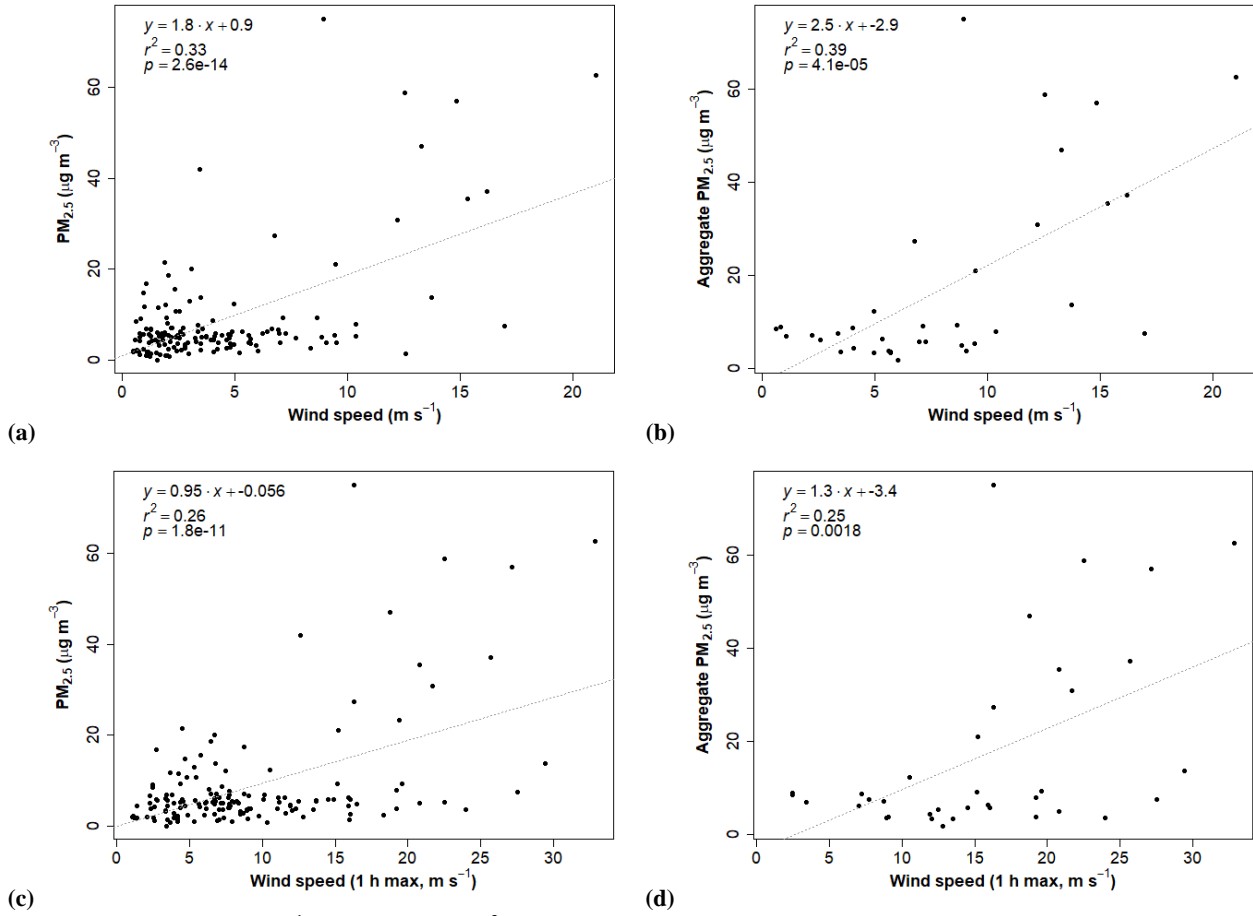

**Figure 12: Wind speed (m s⁻¹) and PM2.5 (µg m⁻³) at PEAV for (a) all data and (b) days with aggregates and for maximum 1 h wind speed for (c) all data and (d) days with aggregates.**





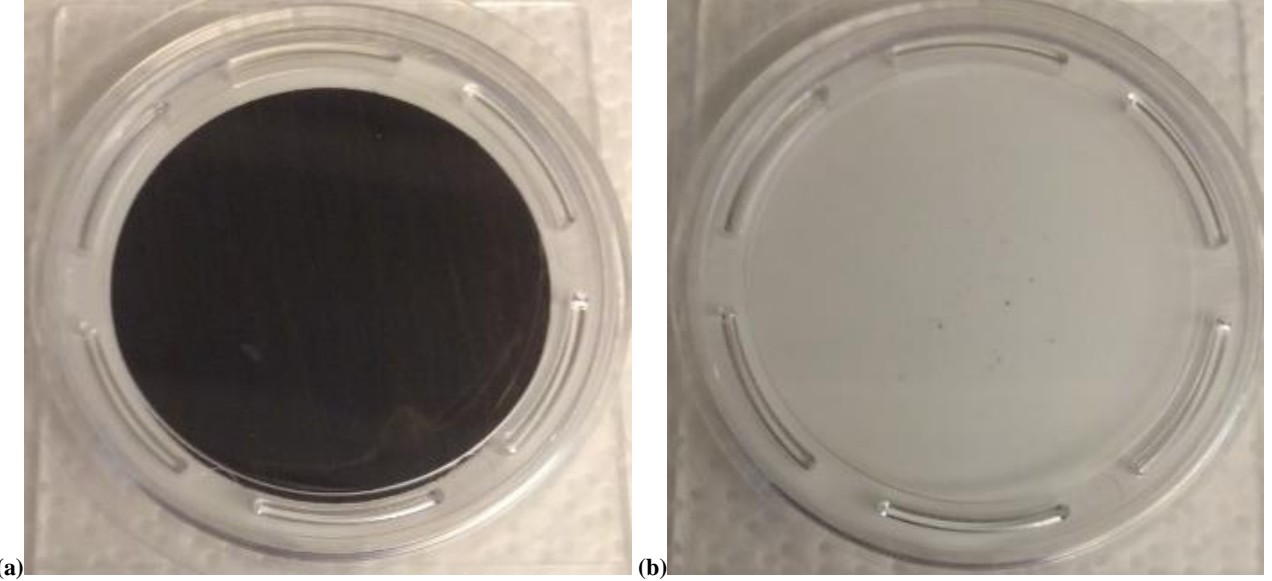

(a)                                                                                    (b)

**Figure 13: Teflon filters from (a) October 21, 2014 during multiple generators running at PEAV and (b) October 25, 2014 the day after multiple generators were running at PEAV.**



**Tables**

| Filter | Inlet | Visual loading | Mass concentration >7.1 µg m⁻³ | Fire? | SEM/ EDS? | Description | Wind speed >20 m s⁻¹ or >10 m s⁻¹ for >10 hours |
|---|---|---|---|---|---|---|---|
| 12-Jun | Both | Medium to heavy | ✓ | | | | >20 m s⁻¹ |
| 13-Jun | CEM | Light | | | | | |
| 16-Jun | CEM | Light | | | | | |
| 19-Jun | CEM | Light | | | | | |
| 25-Jun | Both | Light to medium | | | ✓ | Compact, some fluffy | >10 h |
| 1-Jul | CEM | Light | | ✓ | | | |
| 22-Jul | Both | Light | | ✓ | | | |
| 23-Jul | Both | Light to medium | ✓ | ✓ | ✓ | Compact, some fluffy | >10 h |
| 28-Jul | Teflon | Light | | ✓ | ✓ | Compact, some more diffuse | |
| 12-Aug | Both | Medium | ✓ | | ✓ | Compact and some larger structures | >10 h |
| 13-Aug | Both | Medium to heavy | ✓ | ✓ | | | >20m s⁻¹ |
| 14-Aug | Both | Light | ✓ | ✓ | | | |
| 18-Aug | CEM | Light | | ✓ | | | |
| 19-Aug | CEM | Light | ✓ | ✓ | | | |
| 20-Aug | CEM | Light | ✓ | ✓ | | | |
| 22-Aug | Both | Light | | ✓ | ✓ | Compact and fluffy | |
| 7-Sep | CEM | Light | | ✓ | | | |
| 8-Sep | CEM | Light | | ✓ | | | |
| 11-Sep | Both | Light | ✓ | ✓ | ✓ | Compact, some fluffy | |
| 14-Sep | Both | Light | ✓ | ✓ | | | |
| 15-Sep | Both | Light | ✓ | ✓ | ✓ | Compact and fluffy | |
| 16-Sep | Both | Medium | ✓ | ✓ | ✓ | Compact | >10h, >20m s⁻¹ |
| 17-Sep | Both | Medium | ✓ | ✓ | | | >10h, >20m s⁻¹ |
| 18-Sep | Both | Light | ✓ | ✓ | ✓ | Compact, some fluffy | >10h |
| 20-Sep | CEM | Light | | ✓ | | | |
| 23-Sep | Both | Light to medium | ✓ | ✓ | ✓ | Compact, some fluffy, some spores/pollen | >10h |
| 24-Sep | Both | Medium to heavy | ✓ | ✓ | | | >10h, >20m s⁻¹ |
| 25-Sep | Both | Light to medium | ✓ | | ✓ | Compact, some fluffy | >10h |
| 26-Sep | CEM | Light | | NA | | | |
| 3-Oct | Both | Light | | ✓ | | | |
| 13-Oct | CEM | Light | | ✓ | | | |
| 14-Oct | Both | Heavy | ✓ | NA | ✓ | Compact, disperse | >10h, >20m s⁻¹ |
| 15-Oct | Both | Medium | ✓ | NA | | | >10h, >20m s⁻¹ |
| 19-Oct | Both | Light | | | | | |
| 25-Oct | Both | Light to medium | ✓ | ✓ | | | >10h, >20m s⁻¹ |
| 30-Oct | CEM | Light | | NA | | | >20m s⁻¹ |

**Table 1: Days at PEAV in 2014 with visible aggregates on filters, the inlet with aggregates, and the loading on the filters, whether mass concentration exceeded 75th percentile value for the sample period (7.1 µg m⁻³), presence of fire flag, SEM and EDS analysis, subjective description of the shape, and whether wind speed exceeded 20 m s⁻¹ for at least one hour of the sample day or wind speed exceeded 10 m s⁻¹ for 10 h or longer leading up to or during the sample day.**





| Filter | Inlet | Composition filter with $PM_{2.5}$ not aggregates (range, %) | | | | | | | | | | | |
|--------|-------|------|--------|-------|------|--------|---------|---------|-----------|----------|-----------|----------|--------|
| | | C | O | F | Na | Mg | Al | Si | S | Cl | K | Ca | Fe |
| **Blank** | | 25-41 | 0-8 | 51-75 | | | | | | | | | |
| **25-Jun** | Both | 27-29 | 4.5-5.0 | 65 | | | 2.0-2.2 | 0.15 | | | | | |
| **28-Jul** | Teflon | 29-30 | 3.4-11 | 50-67 | 6.5 | 0.79 | 0.54-2.2 | | 0.18-0.65 | 0.0-0.19 | 0.28 | 0.31 | |
| **12-Aug** | Both | 24-38 | 3.6-8.7 | 56-68 | | | 0.31-3.5 | | 0.21-0.64 | | | | |
| **22-Aug** | Both | 24-37 | 4.6-10 | 58-60 | | | 0.19-6.0 | 0.0-0.14 | 0.13-0.29 | | | | |
| **11-Sep** | Both | 29-39 | 4.7-8.7 | 54-65 | | 0.0-0.14 | 0.26-3.1 | 0.12-0.60 | | | | | |
| **15-Sep** | Both | 44-55 | 7.1-10 | 34-46 | | 0.0-0.14 | 0.0-1.4 | | 0.21-0.75 | | 0.0-0.46 | 0.0-0.79 | |
| **16-Sep** | Both | 35-48 | 13-14 | 35-46 | | | 2.9-5.1 | | 0.33-0.55 | | 0.0-0.27 | | |
| **18-Sep** | Both | 52-53 | 19 | 21-25 | | | 2.4-4.8 | 0.0-2.1 | 0.37-0.57 | | 0.0-0.58 | | |
| **23-Sep** | Both | 45-61 | 12-16 | 23-40 | | | 1.0-5.2 | | 0.0-0.25 | | 0.18-0.19 | | |
| **25-Sep** | Both | 34-29 | 4.7-16 | 49-65 | | 0.0-0.34 | 1.5-5.6 | 0.0-2.1 | 0.0-0.30 | | 0.0-0.62 | | 0.0-3.2 |
| **14-Oct** | Both | 22-26 | 14-18 | 45-53 | | | 8.4-10 | 0.26-0.35 | 0.31-0.42 | | | | |

**Table 2: Elemental composition of sections of Teflon with particulate matter.**



| Filter | Inlet | C | O | F | Mg | Al | Si | S | Cl | K | Ca | Cu |
|--------|-------|---|---|---|-----|-----|-----|------|------|------|------|------|
| **Blank** | | 25-41 | 0-8 | 51-75 | | | | | | | | |
| **25-Jun** | Both | 17-22 | 17-29 | 33-43 | | 18-21 | | 0.0-0.24 | 0.0-0.23 | | | |
| **23-Jul** | Both | 18-28 | 5.2-25 | 37-58 | 0.0-0.32 | 8.3-26 | 0.0-3.3 | 0.0-0.42 | 0.0-0.16 | 0.0-0.89 | | |
| **28-Jul** | Teflon | 20-28 | 8.1-22 | 39-43 | | 14-25 | | 0.19-0.33 | 0.0-0.92 | | | |
| **12-Aug** | Both | 16-26 | 12-34 | 36-52 | 0.0-0.36 | 6.0-17 | | 1.0-3.5 | | | 0.0-0.23 | |
| **22-Aug** | Both | 16-37 | 4.6-28 | 29-59 | | 0.19-29 | | 0.13-0.32 | 0.0-0.29 | | | |
| **11-Sep** | Both | 16-20 | 17-31 | 31-38 | 0.0-0.29 | 17-28 | 0.0-0.37 | 0.0-0.26 | 0.22-0.52 | | | |
| **15-Sep** | Both | 29-34 | 18-26 | 29-32 | | 15-16 | | 0.19-0.26 | | 0.0-0.22 | | |
| **16-Sep** | Both | 20-46 | 11-34 | 15-46 | 0.0-0.24 | 2.2-42 | 0.0-0.23 | 0.0-0.63 | | 0.0-0.25 | | |
| **18-Sep** | Both | 36-60 | 15-38 | 11-30 | 0.0-0.13 | 1.7-16 | | 0.18-0.55 | 0.0-0.38 | 0.0-0.13 | | |
| **23-Sep** | Both | 34-45 | 13-31 | 16-41 | | 1.8-23 | | 0.0-0.26 | 0.0-0.24 | | | |
| **25-Sep** | Both | 18-21 | 21-25 | 39-47 | | 11-17 | | 0.0-0.15 | 0.0-0.44 | | | |
| **14-Oct** | Both | 15-21 | 22-37 | 24-29 | | 22-32 | 0.0-0.37 | 0.26-0.97 | | | | 0.0-0.76 |

**Table 3: Elemental composition of sections of Teflon with visible aggregates.**