# Peer review of "Aggregated particles caused by instrument artifact"

_Atmospheric Measurement Techniques, 2017_

## Referee Comment (RC1) · Anonymous Referee #1 · 19 Dec 2017

This work investigates the observations of superaggregates collected during a study at high and low elevation sites in Nevada in 2014. The authors investigated the potential influence of biomass smoke, a known source of superaggregates. The also investigated other possible sources and deduced that the source of superaggregates was likely related to the anodized Al tubing and fretting corrosion. This work is useful for others investigating observations of superaggregates and the potential role of sampling artifacts. Several fairly minor issues should be addressed before publication as noted in the following comments.

Page 1, Line 17, Can the authors phrase this as 36 out of X sample days?

Page 1, Line 21, This sentence is a bit unclear because it groups the types of influences that could be the source of superaggregates with the types of analysis performed.

Page 1, Line 22, Change "samples with aggregates" to "samples with superaggregates", or state that they are the same thing. I think "aggregates" is used interchange-ably with "superaggregates" throughout the paper, so it might help to be clear about this initially.

Page 1, Line 24, Change "high wind events were the probably reason" to "high wind events were probably the reason"

Page 2, Line 2, Can the authors clarify what they mean by "can be trapped in a high particle volume fraction"? Page 2, Line 6, Can the authors define "length"

Page 2, Line 8, What are the "different behaviors" the authors are referring to here?

Page 2, Line 10, Can the authors define "fractal dimensions"

Page 2, Line 14, Chakrabarty (2014) should be Chakrabarty et al. (2014)

Page 2, Line 21, How abundant are superaggregates? Are they abundant enough to influence estimates of climate forcing?

Page 2, Line 22, It would help to have a segue between this paragraph and the previous discussion.

Page 2, Line 24, To help the reader, the authors could add "a high elevation site" before "Peavine Peak", and a similar "at a lower elevation site ∼12 km southeast in Reno". The elevations may not be needed here since they are again reported in the site description section.

Page 2, Line 27. Can the authors add the total number of days here, so "36 out of X days".

Page 2, Line 31: Do the superaggregates have to conform to this particular descrip-tion?

Page 2, Line 34: Please define "SEM" and "EDS" at first usage.

Page 3, Line 19-21, What is the purpose of this study and how does it fit in with the

2014 study? Also, change "Statuses" to "Status"

Page 3, Line 24, Please add "mass concentration" after PM2.5

Page 3, line 26, Should "<" actually be ">" (particles larger than 2.5 um are being prevented)

Page 4, line 32, Please add "Particles on the Teflon filters"

Page 5, Line 9, How far away from the sample sites was the AERONET site?

Page 5, Line 12, Was AEE calculated using only 2 wavelengths or as a linear fit to several wavelengths?

Page 5, Line 13, Given that the AERONET site is at the campus building, the impacts of urban pollution on AEE could be misinterpreted as biomass smoke if all AEE>1.8 was flagged as fire. How did the authors separate urban influence from biomass smoke?

Page 5, Line 18, It is again unclear why data from the GBNP site are used here since it is a different time period. It would help to explain this earlier.

Page 6, Line 3, Add PM2.5 before "75th percentile". Does this paragraph only refer to 1 inlet?

Page 6, Line 12, Are "black aggregates" different from superaggregates? A similar comment for "black particles" on line 13. Are "black particles" just a general description or refer to "black aggregates"?

Page 6, Line 27: How did the aggregate analysis differ from the fluffy or compact aggregates shown in Figure 3 versus Figure 7? The visual appearance is quite different. Perhaps Figure 3 is actually a superaggregate?

Page 6, Line 27: Did the elemental ratios differ between aggregates and non-aggregate PM2.5? For example, did the Al/O or the Al/Cu differ?

Page 7, Line 7-12, This is somewhat confusing. All data had r2 = 0.33, aggregates only

was r2 = 0.49 and then no aggregates was r2 = 0.58. If the correlation for aggregates only increased, then why did the correlation increase even higher for no aggregates?

Page 7, Line 7-12, The offset in the regressions suggests that when AOD = 0, PM2.5 = 2 ug/m3 or 6 ug/m3 depending on all data or aggregates, respectively? Can the authors comment on this offset?

Page 7, Line 10, A Theil regression would help with this in that it doesn't heavily weight outliers.

Page 7, line 13, Figure 10(a) isn't referred to in the text. I am also concerned that the AEE used here to indicate smoke influence could also be indicating urban pollution. How did the authors account for this?

Page 7, line 15-16, Change to "nor between AEE and PM2.5 on days with aggregates"

Page 7, line 19, Was RH averaged to 24 hours?

Page 7, line 24, If the authors are interested in decreasing the length of the article, I am not sure that Figure 11 is necessary. The discussion of results may be sufficient. Also, it would help to point out that the hypothesis with investigating RH is that hygroscopic effects could have resulted in larger particles. However, the elemental composition and SEM images don't suggest hygroscopic particles.

Page 8, line 9, Why were multiple generators operated?

Page 8, Line 15, Were generators not in use on the weekends?

Page 8, Line 23-25, High aerosol loading in the atmospheric column doesn't necessarily have to indicate the presence of superaggregates. It could just be related to urban pollution.

Page 8, Line 29, The correlation of AOD and AEE may not be higher if there aren't enough superaggregates to affect the total column?

[Figure]

Page 8, Line 30, It's not really clear why the authors included this paragraph? Did they perform some modeling of the hygroscopicity based on composition measurements? Can they tie this discussion back to their data?

Page 9, Line 30-35, Elements are defined here but not when listed on page 6 (24-29). I suggest being consistent or defining them once at first usage.

Page 10, Line 12-13, "RH and AEE were not correlated with aggregate measurements". Can the authors be more specific about the measurements they are referring to? Presence of aggregates? Length? Composition?

Figures Figure 2: The two blue colors were very similar in my version and so difficult to tell the difference.

Figure 3: The superaggregate shown in this figures is very different from Figure 7-which type was more typical?

Figure 9: Instead of "in the valley", can the authors provide a more defined location of the site?

Figure 10: As mentioned earlier, this figure may not be necessary. But if the authors choose to keep it, the legend for "fire" was not pink although the data points were. Also, please provide the wavelengths over which AEE was calculated in the caption. And the same comment regarding "in the valley" also applies for this caption.

Figure 11: As also mentioned, this figure may not be necessary either.

Tables Table 1: Please define "CEM", "SEM", and "EDS" in the caption. Also please include "PM2.5" in "PM2.5 mass concentration exceeded 75th percentile" in the caption.

Table 2: Please include site location in the table, define "both" and "Teflon". Ideally the reader could get the major points without having to read back through the text and "both" or "teflon" might be unclear.

Table 3: Similar comments as for table 2.

[Figure]

---

## Referee Comment (RC2) · Anonymous Referee #2 · 29 Dec 2017

The paper reports a potential artifact during aerosol sampling due to aluminum tubing fretting. I think that making the community aware of this potential artifact is useful; however, I have some reservations regarding the title of the paper, the focus of the abstract/introduction, and the general organization. While the sentences are mostly clearly written, I found the paper confusing, I'll try to explain why next. As it is, I think, the paper (especially considering the current title) might be misinterpreted as if the superaggregates detected in past work, and discussed in the literature, have been erroneously identified as such, while they were just an artifact. In reality, the current paper and findings have little to say about soot superaggregates, in my opinion. The agglomerated particles presented here have, apparently at least, nothing to do with the soot superaggregates reported and discussed in the literature. These aggregates (e.g., from Figure 7) look very different (even just visually) from the soot aggregates reported in Chakrabarty et al., for example; these are composed of a mixture of elements, in-

cluding an abundant amount of aluminum, while the soot superaggregates are mostly composed of carbon. The soot superaggregates morphology (including the nanostructure of their monomers) is clearly defined in literature, while in this paper there is no detailed morphological analysis to compare to. For example, there is no attempt to find the fractal dimension or the monomer size distribution, or their nanostructure. While I still think it is very valuable to make the community aware of the potential presence of these aggregates due to a possibly common sampling artifact, I think the link in the introduction, in the abstract and in the title to soot superaggregates is not clear at best, and deceiving at worst. In addition, while reading the paper for the first time (especially, because of the issues mentioned above), I was not really clear until the very end, where it was going to take me, and I think the main result should be highlighted much earlier on. So, while most of the method and analytical approach and description can be maintained almost as they are, my suggestion would be to: 1. Change the title and get away from the term superaggregate (these are just aggregates), and make it clear what the study is about: an artifact of particles aggregates containing aluminum. 2. Refocus the introduction, onto the real findings of this study, and much less about the "link" (which in my opinion is non-existent) to soot superaggregates. 3. Make the main findings clear early on in the paper; for example, the abstract does not even mention about the aluminum present in these aggregates and the possible tubing fretting corrosion origin, while I think that's the main interesting finding. Instead most of the abstract, since the beginning, focuses on soot superaggregatss, which again, in the end, have nothing to do with what sampled.

Some specific additional comments: - Line 12, page 6: Why "black"? This becomes clear later, but here is not clear. - Line 22-24, page 8: These correlations are discussed earlier on, the repetition here is a bit confusing, I would suggest consolidating all in the discussion. - Paragraph starting at line 30 of page 8: It is not very clear to me why hygroscopic growth is even considered or discussed here. I am not saying it should not be discussed, I am just suggesting it should be made clear why hygroscopic growth should result in aggregation? What is the hypothesis (e.g., a mechanism) behind a

possible link between the hygroscopic growth and the presence/formation of these aggregates? Line 34, page 9 and line 1 on page 10: SEM stays for scanning electron microscopy, I think; so, I believe you should not write "SEM... collected", you can't really collect SEM; maybe "SEM samples... collected" or "SEM images... collected", or something similar?

To summarize, from the point of view of the main material presented here, I would say only minor revisions are needed (no need for new or different analysis or data, for example). I chose major revisions just to underline that a change in title, focus, and organization, would make the paper stronger, clearer and more appropriate.

---

## Author Comment (AC1) · 25 Feb 2018

Anonymous Referee #1 This work investigates the observations of superaggregates collected during a study at high and low elevation sites in Nevada in 2014. The authors investigated the potential influence of biomass smoke, a known source of superaggregates. The also investigated other possible sources and deduced that the source of superaggregates was likely related to the anodized Al tubing and fretting corrosion. This work is useful for others investigating observations of superaggregates and the potential role of sampling artifacts. Several fairly minor issues should be addressed before publication as noted in the following comments.

Response: Thank you for your thorough review of the manuscript.

Page 1, Line 17, Can the authors phrase this as 36 out of X sample days?

[Figure]

Response: This has been changed to "36 out of 158 sample days".

Page 1, Line 21, This sentence is a bit unclear because it groups the types of influences that could be the source of superaggregates with the types of analysis performed.

Response: This sentence has been changed to "To determine if the particles were superaggregates or an instrument artifact, the presence of certain elements, the occurrence of fires, high relative humidity and wind speeds, as well as the use of generators onsite were investigated."

Page 1, Line 22, Change "samples with aggregates" to "samples with superaggregates", or state that they are the same thing. I think "aggregates" is used interchangeably with "superaggregates" throughout the paper, so it might help to be clear about this initially.

Response: The term "aggregates" is used when discussing large aggregated particulates on the filter surfaces under discussion while "superaggregates" is specific to aggregates that have made it past the inlets and are similar in size and morphology to previous studies. This has been clarified throughout the abstract and introduction.

Page 1, Line 24, Change "high wind events were the probably reason" to "high wind events were probably the reason"

Response: "probably" has been changed to "probable"

Page 2, Line 2, Can the authors clarify what they mean by "can be trapped in a high particle volume fraction"?

Response: Large number of particles in a small area causing aggregation. This has not been changed as this is similar terminology to previous studies.

Page 2, Line 6, Can the authors define "length"

Response: Length has been removed.

Page 2, Line 8, What are the "different behaviors" the authors are referring to here?

Response: This has been removed based on comments from referee 2.

Page 2, Line 10, Can the authors define "fractal dimensions"

Response: This has been removed based on comments from referee 2.

Page 2, Line 14, Chakrabarty (2014) should be Chakrabarty et al. (2014)

Response: This has been corrected and moved.

Page 2, Line 21, How abundant are superaggregates? Are they abundant enough to influence estimates of climate forcing?

Response: This has been removed based on comments from referee 2.

Page 2, Line 22, It would help to have a segue between this paragraph and the previous discussion.

Response: This section has changed based on comments from referee 2.

Page 2, Line 24, To help the reader, the authors could add "a high elevation site" before "Peavine Peak", and a similar "at a lower elevation site _12 km southeast in Reno". The elevations may not be needed here since they are again reported in the site description section.

Response: This has been changed.

Page 2, Line 27. Can the authors add the total number of days here, so "36 out of X days".

Response: This has been added.

Page 2, Line 31: Do the superaggregates have to conform to this particular description?

Response: Yes. If the aggregates are not "fluffy" or if the particles are much larger than

they likely have different fractal dimensions and different behavior in regards to inlets. "(e.g. fractal dimensions and lengths)" has been added here.

Page 2, Line 34: Please define "SEM" and "EDS" at first usage.

Response: SEM and EDS were defined on Page 2 lines 12 and 13. However, this section has changed based on comments from referee 2.

Page 3, Line 19-21, What is the purpose of this study and how does it fit in with the 2014 study? Also, change "Statuses" to "Status"

Response: "The TAPI located at PEAV was then moved to GBNP and measurements were made at this high elevation site with impacts from wildfires." was added here for clarification. Statuses has been changed to Status.

Page 3, Line 24, Please add "mass concentration" after PM2.5

Response: This has been added

Page 3, line 26, Should "<" actually be ">" (particles larger than 2.5 um are being prevented)

Response: Yes, thank you for catching that, it has been changed.

Page 4, line 32, Please add "Particles on the Teflon filters"

Response: This has been added.

Page 5, Line 9, How far away from the sample sites was the AERONET site?

Response: Distance has been added here.

Page 5, Line 12, Was AEE calculated using only 2 wavelengths or as a linear fit to several wavelengths?

Response: The wavelengths used for the AEE calculation were provided in the first sentence of this paragraph (440 and 870 nm).

Page 5, Line 13, Given that the AERONET site is at the campus building, the impacts of urban pollution on AEE could be misinterpreted as biomass smoke if all AEE>1.8 was flagged as fire. How did the authors separate urban influence from biomass smoke?

Response: Previous studies cited Loría-Salazar et al. 2014, 2016, and 2017 discuss the use of the fire flags. These studies conclude that local conditions of aerosol pollution in Reno, Nevada is most likely impacted by coarse mode particles when AEE is lower than 1.2. Urban emissions of aerosol pollution have an AEE that ranges from ~1.2 to ~1.7. Fine mode fraction coming from fires shows very high fine mode fraction and AEE larger than 1.8. "...and when fine mode fraction was > 0.6..." was added here for further clarity.

Page 5, Line 18, It is again unclear why data from the GBNP site are used here since it is a different time period. It would help to explain this earlier.

Response: A sentence was added to the site description section.

Page 6, Line 3, Add PM2.5 before "75th percentile". Does this paragraph only refer to 1 inlet?

Response: PM2.5 was added before concentration here.

Page 6, Line 12, Are "black aggregates" different from superaggregates? A similar comment for "black particles" on line 13. Are "black particles" just a general description or refer to "black aggregates"?

Response: Black aggregates refer to aggregates under investigation here and are not necessarily superaggregates as defined in previous work. "Black particles" has been changed to "black aggregates" here.

Page 6, Line 27: How did the aggregate analysis differ from the fluffy or compact aggregates shown in Figure 3 versus Figure 7? The visual appearance is quite different. Perhaps Figure 3 is actually a superaggregate?

Response: Analysis for fluffy vs. compact aggregates was the same, SEM and EDS. The first SEM images were a cursory look at the aggregates; the second SEM analysis was more thorough. Yes, the goal of Fig. 3 and 7 is to illustrate that it is possible to find aggregates that look like superaggregates. Fluffy aggregates such as Fig. 3 lead to the hypothesis that superaggregates may be depositing past the inlets, however, these are not the particles that are visibly black and dominant on the filters as they are much smaller and not common. This is explained in the previous paragraph on this page and in section 4.2.

Page 6, Line 27: Did the elemental ratios differ between aggregates and non-aggregate PM2.5? For example, did the Al/O or the Al/Cu differ?

Response: The Al:O ratio differed from aggregates (0.40 to 0.91) to non-aggregates (0.06 to 0.61). Cu only occurred on one filter and only in an aggregate. Al:C and Al:F also differed between aggregate and non-aggregate. The rest of the elements were not consistent throughout the samples but differed when present on aggregates and non-aggregate samples. The authors think that this information is illustrated by the differences in the percentages of the elements.

Page 7, Line 7-12, This is somewhat confusing. All data had r2 = 0.33, aggregates only was r2 = 0.49 and then no aggregates was r2 = 0.58. If the correlation for aggregates only increased, then why did the correlation increase even higher for no aggregates?

Response: The aggregate only and the non-aggregate datasets are differently correlated, when combined into one dataset the correlation for all data decreased.

Page 7, Line 7-12, The offset in the regressions suggests that when AOD = 0, PM2.5 = 2 ug/m3 or 6 ug/m3 depending on all data or aggregates, respectively? Can the authors comment on this offset?

Response: AOD is a measure of the columnar aerosol loading and does not necessarily directly relate to the surface PM2.5 measurements. Both instruments also have

different minimum detection limits that affect the offset.

Page 7, Line 10, A Theil regression would help with this in that it doesn't heavily weight outliers.

Response: Thank you for the suggestion. A Theil regression would be helpful if there were multiple outliers and if AOD was a probable cause of the aggregates. However, because there is only one major outlier and AOD is not the probable cause, we do not think it would add more insight to the paper.

Page 7, line 13, Figure 10(a) isn't referred to in the text. I am also concerned that the AEE used here to indicate smoke influence could also be indicating urban pollution. How did the authors account for this?

Response: 10a has been added here. Previous studies cited in section 3.3 Loría-Salazar et al. 2014, 2016, and 2017 discuss the use of the fire flags, clarification has been added in section 3.3.

Page 7, line 15-16, Change to "nor between AEE and PM2.5 on days with aggregates"

Response: This has been changed.

Page 7, line 19, Was RH averaged to 24 hours?

Response: Yes, "24 h" has been added to this line.

Page 7, line 24, If the authors are interested in decreasing the length of the article, I am not sure that Figure 11 is necessary. The discussion of results may be sufficient. Also, it would help to point out that the hypothesis with investigating RH is that hygroscopic effects could have resulted in larger particles. However, the elemental composition and SEM images don't suggest hygroscopic particles.

Response: Figure 11 has been removed. The last sentence of this paragraph was modified to indicate that hygroscopic growth could be causing larger particles.

Page 8, line 9, Why were multiple generators operated?

Response: This is explained in section 3.5.

Page 8, Line 15, Were generators not in use on the weekends?

Response: We do not know. Regular maintenance of anything at the station would occur during normal work hours unless there was an unexpected loss of power, which we are not aware of occurring during the sample period. This has been clarified.

Page 8, Line 23-25, High aerosol loading in the atmospheric column doesn't necessarily have to indicate the presence of superaggregates. It could just be related to urban pollution.

Response: This sentence points out that the correlation between high column loading and aggregates suggested that there could be a relationship between high column loading and aggregates, not that it indicates the presence of superaggregates.

Page 8, Line 29, The correlation of AOD and AEE may not be higher if there aren't enough superaggregates to affect the total column?

Response: This is based on findings in Loría-Salazar et al. 2017, which identifies the atmospheric physical conditions where AOD is associated with PM2.5.

Page 8, Line 30, It's not really clear why the authors included this paragraph? Did they perform some modeling of the hygroscopicity based on composition measurements? Can they tie this discussion back to their data?

Response: This paragraph explains what % RH is important for determining hygroscopic growth of different particulates. Some clarification has been added to this and the following paragraph.

Page 9, Line 30-35, Elements are defined here but not when listed on page 6 (24-29). I suggest being consistent or defining them once at first usage.

Response: This has been fixed.

Page 10, Line 12-13, "RH and AEE were not correlated with aggregate measurements". Can the authors be more specific about the measurements they are referring to? Presence of aggregates? Length? Composition?

Response: This has been changed to presence of aggregates.

Figures Figure 2: The two blue colors were very similar in my version and so difficult to tell the difference.

Response: The blues have been changed to increase visibility.

Figure 3: The superaggregate shown in this figures is very different from Figure 7-which type was more typical?

Response: This is discussed in the manuscript in the introduction (section 1), section 4.2, and table 1.

Figure 9: Instead of "in the valley", can the authors provide a more defined location of the site?

Response: This has been changed to "near the low elevation site in Reno".

Figure 10: As mentioned earlier, this figure may not be necessary. But if the authors choose to keep it, the legend for "fire" was not pink although the data points were. Also, please provide the wavelengths over which AEE was calculated in the caption. And the same comment regarding "in the valley" also applies for this caption.

Response: This figure has been removed.

Figure 11: As also mentioned, this figure may not be necessary either.

Response: This figure has been removed.

Tables Table 1: Please define "CEM", "SEM", and "EDS" in the caption. Also please include "PM2.5" in "PM2.5 mass concentration exceeded 75th percentile" in the caption.

Response: This has been added.

Table 2: Please include site location in the table, define "both" and "Teflon". Ideally the reader could get the major points without having to read back through the text and "both" or "teflon" might be unclear.

Response: This has been added.

Table 3: Similar comments as for table 2.

Response: This has been added.

---

## Author Comment (AC2) · 25 Feb 2018

Anonymous Referee #2 The paper reports a potential artifact during aerosol sampling due to aluminum tubing fretting. I think that making the community aware of this potential artifact is useful; however, I have some reservations regarding the title of the paper, the focus of the abstract/introduction, and the general organization. While the sentences are mostly clearly written, I found the paper confusing, I'll try to explain why next. As it is, I think, the paper (especially considering the current title) might be misinterpreted as if the superaggregates detected in past work, and discussed in the literature, have been erroneously identified as such, while they were just an artifact. In reality, the current paper and findings have little to say about soot superaggregates, in my opinion. The agglomerated particles presented here have, apparently at least, nothing to do with the soot superaggregates reported and discussed in the literature. These aggregates (e.g., from Figure 7) look

very different (even just visually) from the soot aggregates reported in Chakrabarty et al., for example; these are composed of a mixture of elements, in- cluding an abundant amount of aluminum, while the soot superaggregates are mostly composed of carbon. The soot superaggregates morphology (including the nanostructure of their monomers) is clearly defined in literature, while in this paper there is no detailed morphological analysis to compare to. For example, there is no attempt to find the fractal dimension or the monomer size distribution, or their nanostructure. While I still think it is very valuable to make the community aware of the potential presence of these aggregates due to a possibly common sampling artifact, I think the link in the introduction, in the abstract and in the title to soot superaggregates is not clear at best, and deceiving at worst. In addition, while reading the paper for the first time (especially, because of the issues mentioned above), I was not really clear until the very end, where it was going to take me, and I think the main result should be highlighted much earlier on. So, while most of the method and analytical approach and description can be maintained almost as they are, my suggestion would be to:

Response: Thank you for your comments on the manuscript. The authors would have liked to perform morphological analysis had there been resources available for such analysis. As it is, the visual inspection and EDS reports, as referee 2 has mentioned, were more than sufficient to determine that the aggregated particles in this study are indeed different from superaggregates in previous studies. It was stated both in the abstract ("However, further analysis revealed that these particles were dissimilar to superaggregates observed in previous studies.") and in the introduction ("The observation of aggregates that did not conform to the description of superaggregates from previous studies led us to wonder if the observed aggregates may in fact be an artifact of the instrument setup and not an ambient air phenomenon.") that these aggregates did not conform to superaggregates previously observed and were likely an instrument artifact and not an ambient air phenomenon. The authors have shortened much of the discussion on superaggregates to reduce any confusion on the intent of the paper.

1. Change the title and get away from the term superaggregate (these are just aggregates), and make it clear what the study is about: an artifact of particles aggregates containing aluminum.

Response: The title has been changed. Based on comments from referee 1, we have clarified that aggregates in this paper refer to large aggregated particles observed in this study while superaggregates refer to large aggregated particles observed in previous studies.

2. Refocus the introduction, onto the real findings of this study, and much less about the "link" (which in my opinion is non-existent) to soot superaggregates.

Response: Due to the finding of aggregates particles that resemble superaggregates, the authors think that it is important to point out caution when analyzing samples as possible "superaggregates". We have shortened much of the introduction related to superaggregates but feel some discussion is important.

3. Make the main findings clear early on in the paper; for example, the abstract does not even mention about the aluminum present in these aggregates and the possible tubing fretting corrosion origin, while I think that's the main interesting finding. Instead most of the abstract, since the beginning, focuses on soot superaggregatss, which again, in the end, have nothing to do with what sampled.

Response: Some discussion on superaggregates has been removed from the abstract and more focus on the differences has been added.

Some specific additional comments: - Line 12, page 6: Why "black"? This becomes clear later, but here is not clear.

Response: This has been changed to "Aggregates, black in color,...)

- Line 22-24, page 8: These correlations are discussed earlier on, the repetition here is a bit confusing, I would suggest consolidating all in the discussion.

Response: The correlations are discussed in the results section in full. The correlations are mentioned again during the discussion to reduce the need to look back at the results when discussing what the correlations indicate. This has not been changed.

- Paragraph starting at line 30 of page 8: It is not very clear to me why hygroscopic growth is even considered or discussed here. I am not saying it should not be discussed, I am just suggesting it should be made clear why hygroscopic growth should result in aggregation? What is the hypothesis (e.g., a mechanism) behind a possible link between the hygroscopic growth and the presence/formation of these aggregates?

Response: A sentence has been added here to clarify this.

Line 34, page 9 and line 1 on page 10: SEM stays for scanning electron microscopy, I think; so, I believe you should not write "SEM... collected", you can't really collect SEM; maybe "SEM samples... collected" or "SEM images... collected", or something similar?

Response: This has been changed to SEM images.

To summarize, from the point of view of the main material presented here, I would say only minor revisions are needed (no need for new or different analysis or data, for example). I chose major revisions just to underline that a change in title, focus, and organization, would make the paper stronger, clearer and more appropriate.